Compacting and correcting Trinity and Oases RNA-Seq de novo assemblies

Cabau Cédric Cedric.Cabau@inra.fr 1
Escudié Frédéric 2
Djari Anis 3
Guiguen Yann 4
Bobe Julien 4
Klopp Christophe 1 2
1 SIGENAE, GenPhySE, Université de Toulouse, INRA, INPT, ENV , Castanet Tolosan , France
2 Plate-forme bio-informatique Genotoul, Mathématiques et Informatique Appliquées de Toulouse, INRA , Castanet Tolosan , France
3 Laboratoire Génomique et Biotechnologie du Fruit, UMR990 INRA/INP-ENSAT , Auzeville , France
4 UR1037 Fish Physiology and Genomics, INRA , Rennes , France
McHardy Alice
Electronic publication date: 2017 Feb 16
Publication date: 2017
Volume: 5
Electronic Location ID: e2988
Received 2016 Jul 11; Accepted 2017 Jan 10
Copyright: ©2017 Cabau et al.
Copyright year: 2017
Copyright holder: Cabau et al.
License: This is an open access article distributed under the terms of the Creative Commons Attribution License, which permits unrestricted use, distribution, reproduction and adaptation in any medium and for any purpose provided that it is properly attributed. For attribution, the original author(s), title, publication source (PeerJ) and either DOI or URL of the article must be cited.
License URL: https://creativecommons.org/licenses/by/4.0/

Keywords: RNA-Seq, De novo assembly, Compaction, Correction, Quality assessment

Funding: PhyloFish ANR-10-GENM-017 France Génomique ANR10-INBS-09-08 This work was supported by funds from the ANR (http://www.agence-nationale-recherche.fr/) in the frame of the PhyloFish (ANR-10-GENM-017) and the France Génomique (ANR10-INBS-09-08) projects. The funders had no role in study design, data collection and analysis, decision to publish, or preparation of the manuscript.

==============================
Background

De novo transcriptome assembly of short reads is now a common step in expression analysis of organisms lacking a reference genome sequence. Several software packages are available to perform this task. Even if their results are of good quality it is still possible to improve them in several ways including redundancy reduction or error correction. Trinity and Oases are two commonly used de novo transcriptome assemblers. The contig sets they produce are of good quality. Still, their compaction (number of contigs needed to represent the transcriptome) and their quality (chimera and nucleotide error rates) can be improved.

Results

We built a de novo RNA-Seq Assembly Pipeline (DRAP) which wraps these two assemblers (Trinity and Oases) in order to improve their results regarding the above-mentioned criteria. DRAP reduces from 1.3 to 15 fold the number of resulting contigs of the assemblies depending on the read set and the assembler used. This article presents seven assembly comparisons showing in some cases drastic improvements when using DRAP. DRAP does not significantly impair assembly quality metrics such are read realignment rate or protein reconstruction counts.

Conclusion

Transcriptome assembly is a challenging computational task even if good solutions are already available to end-users, these solutions can still be improved while conserving the overall representation and quality of the assembly. The de novo RNA-Seq Assembly Pipeline (DRAP) is an easy to use software package to produce compact and corrected transcript set. DRAP is free, open-source and available under GPL V3 license at http://www.sigenae.org/drap.

Background

Second-generation sequencing platforms have enabled the production of large amounts of transcriptomic data permitting to analyze gene expression for a large variety of species and conditions. For species lacking a reference genome sequence, the now-classical processing pipeline includes a de novo transcriptome assembly step. Assembling an accurate transcriptome reference is difficult because of the raw data variability. This variability comes from different factors: including: 1. The variability of gene expression levels ranging usually between one and millions of copies; 2. The biology of mRNA synthesis which goes through an early stage of pre-mRNA still containing introns and a late state in which mRNA can be decayed; 3. The synthesis from pre-mRNA of numerous alternative transcripts; 4. Potential sample contaminations; 5. Sequencing quality biases; 6. Most of the genome can be expressed in low abundance depending on the biological condition as presented by Djebali et al. (2012) in the results of the ENCODE project.

Today there is no unique best solution to these RNA-Seq assembly problems but several software packages have been proven to generate contig sets comprising most of the expressed transcripts correctly reconstructed. Trinity (Grabherr et al., 2011) and Oases (Schulz et al., 2012) are good examples. The assembled contig sets produced by these packages often contain multiple copies of complete or partial transcripts and also chimeras. Chimeras are structural anomalies of a unique transcript (self-chimeras) or multiple transcripts (multi-transcripts chimeras). They are called “cis” if the transcripts are in the same direction and “trans” if they are in opposite directions. Natural chimeric transcripts exist in some cancer tissues but are rare (Frenkel-Morgenstern et al., 2013). Yang & Smith (2013) have shown the tendency of de novo transcriptome assemblers to produce self-chimeric contigs. The prevalence of the phenomenon depends on the assembly parameters. Multi-transcript chimeras distort contig annotation. The functions of the transcripts merged in the same contig can be very different and therefore the often-unique annotation given to such a chimeric contig does not reflect its content. Assemblies include also contigs corresponding to transcription or sequencing noise a phenomenon often referred as illegitimate transcription (Chelly et al., 1989). These contigs have often low coverage and are not found in the different replicates of the same condition.

Some contigs contain local biological variations or sequencing errors such as substitutions, insertions or deletions. These variations and errors can deeply impact the read alignment rate, create frameshifts which hinder annotation, limit the efficacy of primer design and generate false variations. Assemblies contain also polyA/T tails, which are posttranscriptional marks. They are usually removed before publication. For all these reasons contig sets usually need error correction.

Trinity and Oases have different algorithms, which give them advantages or disadvantages depending on gene expression levels. The main difference comes from their assembly strategy. Trinity chains a greedy algorithm with a de Bruijn graph one and Oases uses multiple de Bruijn graphs with different k-mers. The first step of Trinity is very effective in assembling parts of highly expressed transcripts which will be connected at the second step. As shown by Surget-Groba & Montoya-Burgos (2010), the Oases multi-k assembly approach is able to build contigs corresponding to transcripts with very low to very high expression levels. However, highly expressed genes with multiple transcripts will generate very complex graphs mainly because of the presence of variations or sequencing errors, which will form new paths possibly considered as valid by the assembler and produce numerous erroneous contigs. No assembler is producing the best contig set in all situations. Bio-informaticians and biologists therefore use different strategies to maximize the reference contig set quality (Mbandi et al., 2015; Bens et al., 2016; He et al., 2015; Nakasugi et al., 2014). The simplest approach is to produce a reference set per software package or parameter set, to compare their metrics and choose the best one. It is also possible to merge different results and filter them.

Assemblies can be compared on different criteria. The usual ones are simple contig metrics such as total count, total length, N50, and average length. Assembling equals summarizing (compressing the expression dimension) and therefore a good metric to check the summary quality is the proportion of reads mapped back to the contigs. As a large part of the transcripts correspond to mRNA, it is also possible to use as quality metric the number of correctly reconstructed proteins using a global reference as it is done by CEGMA (Parra, Bradnam & Korf, 2007) or BUSCO (Simão et al., 2015) or using a protein reference set from a phylogenetically closely related organism. Last, some software packages are also rating the contig set or the individual contigs using the above-mentioned criteria (Honaas et al., 2016) or some other for example only related to the way reads map back to the contigs (Smith-Unna et al., 2016; Li et al., 2014; Davidson & Oshlack, 2014).

We have built a de novo RNA-Seq Assembly Pipeline (DRAP) in order to correct the following assembly problems: multiple copies of complete or partial transcripts, chimeras, lowly expressed intergenic transcription, insertion and deletion generated by the assemblers and polyA tails. The pipeline implementation is presented in the next section. The “results and discussion” section compares raw and DRAP assembly metrics for seven different datasets.

Implementation

DRAP is written in Perl, Python, and shell. The software is a set of three command-line tools respectively called runDrap, runMeta and runAssessment. runDrap performs the assembly including compaction and correction. It produces a contig set but also a HTML log report presenting different assembly metrics. runAssessment compares different contig sets and gathers the results in a global report. runMeta merges and compacts different contigs sets and should be used for very large datasets for which memory or CPU requirements do not enable a unique global assembly or for highly complex datasets. The modules chained by each tool are presented in a graphical manner in Figs. 1, 2 and 3. Details on the compaction, correction and quality assessment steps of the tools are described hereafter. All software versions, parameters and corresponding default values are presented in Table S1.

Figure 1 Steps in runDRAP workflow.

This workflow is used to produce an assembly from one sample/tissue/development stage. It take as input R1 from single-end sequencing or R1 and R2 from paired-end sequencing and eventually a reference proteins set from closest species with known proteins.

Figure 2 Steps in runMeta workflow.

This workflow is used to produce a merged assembly from several samples/tissues/development stage outputted by runDRAP. Inputs are runDRAP output folders and eventually a reference protein set.

Figure 3 Steps in runAssessment workflow.

This workflow is used to evaluate quality for one assembly or for compare several assemblies produced from the same dataset. Inputs are the assembly/ies, R1 and eventually R2, and a reference protein set.

Contig set compaction

Contig compaction removes redundant and lowly expressed contigs. Four different approaches are used to compact contig sets. The first is only implemented for Oases assemblies and corresponds to the sub-selection of only one contig per locus (NODE) produced by the assembler. Oases resolves the connected component of the de Bruijn graph and for complex sub-graphs generates several longest paths corresponding to different possible forms. These forms have shown (https://sites.google.com/a/brown.edu/bioinformatics-in-biomed/velvet-and-oases-transcriptome) to correspond to subpart of the same transcript, which are usually included one in another. Oases provides the locus (connected component of the assembly graph) of origin of each contig as well as its length and depth. The Oasesv2.0.4BestTransChooser.py script sub-selects the longest and most covered contig of a locus. The second compaction method removes contigs included in longer ones. CD-HIT-EST (Fu et al., 2012) orders the contigs by length and removes all the included ones given identity and coverage thresholds. The third method elongates the contigs through a new assembly step. TGICL (Pertea et al., 2003) performs this assembly in DRAP. The last approach filters contigs using their length or the length of their longest ORF if users are only interested in coding transcripts, and using read coverage according to the idea that lowly covered contigs often correspond to noise. A last optional filter selects contigs using their TransRate quality score when above the calculated threshold (–optimize parameter). By default, runDrap produces eight contigs sets, four include only protein coding transcripts and four others contain all transcripts. Each group comprises a contig set filtered for low coverage with respectively 1, 3, 5 and 10 fragments per kilobase per million (FPKM) thresholds.

Compaction favors assemblies having contigs with multiple ORFs. Because a unique ORF is expected for contig annotation, DRAP splits multi-transcript chimera in mono-ORF contigs.

runMeta also performs a three step compaction of the contigs. The first is based on the contig nucleotide content and uses CD-HIT-EST. The second run CD-HIT on the protein translation of the longest ORF found by EMBOSS getorf. The third, in the same way as runDrap, filters contigs using their length (global or longest ORF), their expression level and optionally their TransRate score producing the eight result files described in the previous paragraph.

Contig set corrections

Contig correction splits chimeras, removes duplicated parts, removes insertions, deletion and polyA/T tails. DRAP corrects contigs in three ways. It first searches self-chimera and removes them by splitting contigs in parts or removing duplicated chimeric elements. An in house script aligns contigs on themselves using bl2seq and keeps only matches having an identity greater or equal to 96%. A contig is defined as a putative chimera if (i) the longest self-match covers at least 60% of the contig length or (ii) the sum of partial non-overlapping self-matches covers at least 80% of its length. In the first case, the putative chimera is split at the start position of the repeated block. In the second case, the contig is only a repetition of a short single block and is therefore discarded. For the second correction step, DRAP searches substitutions, insertions and deletions in the read realignment file. When found it corrects the consensus according to the most represented allele at a given position. Low read coverage alignment areas are usually not very informative therefore only positions having a minimum depth of 10 reads are corrected. The manual assessment made on DRAP assemblies has shown that a second path of this algorithm improves consensus correction. Part of the reads change alignment location after the first correction. runDrap, consequently, runs this step twice.

The last correction script eases the publication of the contig set in TSA (https://www.ncbi.nlm.nih.gov/genbank/tsa): NCBI transcript sequence assembly archive. TSA stores the de novo assembled contig sets of over 1300 projects. In order to improve the data quality, it performs several tests before accepting a new submission. These tests search for different elements such as sequencing adapters or vectors, polyA or polyT and stretches of unknown nucleotides (N). The thresholds used by TSA are presented at https://www.ncbi.nlm.nih.gov/genbank/tsaguide. DRAP performs the same searches on the contig set and corrects the contigs when needed.

Quality assessment

All three workflows create an HTML report. The report is a template including HighCharts (http://www.highcharts.com) graphics and tables using JSON files as database. These files are generated by the different processing steps. The report can therefore also be used to monitor processing progression. Each graphic included in the report can be downloaded in PNG, GIF, PDF or SVG. Some of the graphics can be zoomed in by mouse selecting the area to be enlarged. The report tables can be sorted by clicking on the column headers and exported in CSV format. For runDrap and runMeta, the reports present results of a single contig file.

runAssessment processes one or several contig files and one or several read files. It calculates classical contig metrics, checks for chimeras, searches alignment discrepancies, produces read and fragment alignment rates and assess completeness using an external global reference running BUSCO. If provided, it aligns a set of proteins on the contigs to measure their overlap. Last, it runs TransRate, a contig validation software using four alignment linked quality measures to generate a global quality criterion for each contig and for the complete set. runAssessment does not modify the contig set content but enables users to check and select the best candidate between different assemblies.

Parallel processing and flow control

DRAP runs on Unix machines or clusters. Different steps of the assembly or assessment process are run in parallel mode, if the needed computer infrastructure is available. All modules have been implemented to take advantage of an SGE compliant HPC environment. They can be adapted to other schedulers through configuration file modification.

DRAP first creates a set of directories and shell command files and then launches these files in the predefined order. The ‘–write’ command line parameter forces DRAP to stop after the first step. At this stage, the user can modify the command files for example to set parameters which are not directly accessible from runDRAP, runMeta or runAssessment and then launch the process with the ‘–run’ command line option.

DRAP checks execution outputs at each processing step. If an error has occurred, it adds an error file to the output directory indicating at which step of the processing it happened. After correction, DRAP can be launched again and it will scan the result directory and restart after the last error free step. The pipeline can easily be modified to accept other assemblers by rewriting the corresponding wrapper using the input files and producing correctly named output files.

Table 1 Datasets.

Name	Species	Layout	Library	Protocol	
		Paired	Stranded	Length (nt)	Nb R1	SRA ID	Tissue	Condition	
At	Arabidopsis thaliana	Yes	–	100	32,041,730	SRR1773557	Root	Full nutrition	
		Yes	–	100	30,990,531	SRR1773560	Shoot	Full nutrition	
		Yes	–	100	24,898,527	SRR1773563	Root	N starvation	
		Yes	–	100	54,344,171	SRR1773569	Flower	Full nutrition	
		Yes	–	150	31,467,967	SRR1773580	Shoot	N starvation	
Bt	Bos taurus	Yes	No	100	30,140,101	SRR2635009	Milk	Day 70 with low milk production	
		Yes	No	75	15,339,206	SRR2659964	Endometrium	–	
		Yes	Yes	50	13,542,516	SRR2891058	Oviduct	–	
Dd	Danio rerio	Yes	No	100	35,368,936	SRR1524238	Brain	5 months female	
					54,472,116	SRR1524239	Gills	5 months female	
					85,672,616	SRR1524240	Heart	5 months male and female	
					34,032,976	SRR1524241	Muscle	5 months female	
					59,248,034	SRR1524242	Liver	5 months female	
					46,371,614	SRR1524243	Kidney	5 months male and female	
					96,715,965	SRR1524244	Bones	5 months female	
					43,187,341	SRR1524245	Intestine	5 months female	
					55,185,501	SRR1524246	Embryo	2 days embryo	
					24,878,233	SRR1524247	Unfertilized eggs	5 months female	
					22,026,486	SRR1524248	Ovary	5 months female	
					59,897,686	SRR1524249	Testis	5 months male	
Dm	Drosophila melanogaster	Yes	Yes	75	21,849,652	SRR2496909	Cell line R4	Time P17	
					21,864,887	SRR2496910	Cell line R4	Time P19	
					20,194,362	SRR2496918	Cell line R5	Time P17	
					22,596,303	SRR2496919	Cell line R5	Time P19	
Dr	Danio rerio	Yes	No	100	5,072,822	SRR1048059	Pineal gland	Light	
					8,451,113	SRR1048060	Pineal gland	Light	
					8,753,789	SRR1048061	Pineal gland	Dark	
					7,420,748	SRR1048062	Pineal gland	Dark	
					9,737,614	SRR1048063	Pineal gland	Dark	
Ds	Danio rerio	Yes	No	100	30,000,000	Simulated	–	–	
Hs	Homo sapiens	No	No	25–50	15,885,224	SRR2569874	TK6 cells	pretreated with the protein kinase C activating tumor	
					15,133,619	SRR2569875	TK6 cells	pretreated with the protein kinase C activating tumor	
					19,312,543	SRR2569877	TK6 cells	pretreated with the protein kinase C activating tumor	
					21,956,840	SRR2569878	TK6 cells	pretreated with the protein kinase C activating tumor	

Results and Discussion

DRAP has been tested on seven different datasets corresponding to five species. These datasets are presented in Table 1 and include five real datasets (Arabidopsis thaliana: At, Bos taurus: Bt, Drosophila melanogaster: Dm, Danio rerio: Dr and Homo sapiens: Hs), one set comprising a large number of diverse samples (Danio rerio multi samples: Dd) and one simulated dataset (Danio rerio simulated: Ds). The simulated reads have been produced using rsem-simulate-reads (version rsem-1.2.18) (Li & Dewey, 2011). The theta0 value was calculated with the rsem-calculate-expression program on read files from the Danio rerio pineal gland sample (SRR1048059). Table 1 also presents for each dataset: the number, length, type (paired or not) and strandedness of the reads, the public accession number, the tissue and experimental condition of origin. The results presented hereafter compare the metrics collected from Trinity, Oases, DRAP Trinity and DRAP Oases assemblies of the six first datasets. The multi sample dataset has been used to compare a strategy in which all reads of the different samples are gathered and processed as one dataset (pooled) to a strategy in which the assemblies are performed by sample and the resulting contigs joined afterwards (meta-assembly). The same assembly pipeline has been used in both strategies, except the contig set merging step, which is specific to the meta-assembly strategy.

Summary Table 2 and Table 3 present the metrics collected for the six first datasets. Table 2 provides metrics related to compaction and correction as Table 3 includes validation metrics and Table 4 collects all three metric types for pooled versus meta-assembly strategies.

Table 2 Compaction and correction in DRAP and standard assembler.

Dataset	Assembler	Nb contigs	N50 (nt)	L50 (nt)	Sum(nt)	Median length (nt)	Included contigs (%)	Contigs with multi-ORF (%)	Contigs with Multi-prot (%)	Chimeric Contigs (%)	Contigs with Bias* (%)	
At	Oases	381,440	2,971	92,020	843,329,264	1,816	72.75	27.89	0.26	0.80	13.88	
	DRAP Oases	32,269	2,014	9,563	56,122,047	1,547	0.00	0.24	1.40	0.04	2.78	
	Trinity	95,008	2,198	19,140	130,969,737	991	4.05	15.63	1.22	0.20	11.29	
	DRAP Trinity	54,923	1,761	15,857	80,258,659	1,287	0.00	0.20	0.52	0.00	2.68	
Bt	Oases	147,163	2,739	31,441	269,085,141	1,359	71.19	7.45	0.06	0.66	6.29	
	DRAP Oases	29,685	2 441	6 029	47,727,730	1 111	0.00	0.28	0.32	0.03	1.23	
	Trinity	89,520	2,184	12,080	90,989,611	431	4.12	3.69	0.17	0.12	5.98	
	DRAP Trinity	46,561	2,129	9,183	64,809,448	927	0.00	0.23	0.14	0.00	1.50	
Dm	Oases	178,696	2,220	29,086	232 776 717	756	75.48	5.14	0.18	0.35	13.11	
	DRAP Oases	21,550	2,309	3,674	29,372,261	804	0.00	0.09	0.45	0.06	2.27	
	Trinity	55,214	2,266	7,126	57,209,890	438	5.19	4.58	0.95	0.22	13.33	
	DRAP Trinity	27,236	2 146	5 240	37,249,612	914	0.00	0.07	0.31	0.00	3.59	
Dr	Oases	702,640	2,715	114,042	1,059,904,844	857	70.99	2.80	0.01	1.39	11.52	
	DRAP Oases	46,831	2,757	9,046	82,268,872	1,173	0.00	0.15	0.27	0.16	13.05	
	Trinity	126,210	1 279	21,003	96,279,046	418	5.56	0.81	0.08	0.56	23.63	
	DRAP Trinity	58,114	1,644	13 022	68,900,396	866	0.00	0.07	0.12	0.00	7.41	
Ds	Oases	131,982	2,975	28,618	280,469,694	1 619	75.05	3.05	0.06	0.14	4.07	
	DRAP Oases	21 191	3 000	4,872	46,994,928	1,744	0.00	0.08	0.25	0.02	1.10	
	Trinity	40,335	2,398	7,159	58,571,859	910	3.12	1.82	0.37	0.09	6.47	
	DRAP Trinity	31,113	2,381	6,492	51,580,407	1,205	0.00	0.04	0.14	0.00	1.15	
Hs	Oases	101,271	2,048	20,131	132,681,065	895	55.73	5.55	0.03	0.11	7.51	
	DRAP Oases	30,201	1,880	5,542	34,670,862	540	0.00	0.15	0.08	0.00	0.68	
	Trinity	57,195	1,687	7,843	47,639,190	384	2.63	2.85	0.12	0.09	5.79	
	DRAP Trinity	39,489	1,705	6,621	38,557,758	540	0.00	0.11	0.06	0.00	0.59	
Notes.

* Contigs with consensus variations corrected by DRAP.

Bold values are “best in class” values between raw and DRAP assemblies.

Table 3 Validation DRAP against standard assembler.

Dataset	Assembler	% contigs by ORF count	Contigs with Complete ORF (%)	% contigs by Proteins count	Nb reference Proteins aligned	Reads mapping (%)	TransRate score * 100	
		0	1		0	1		Mapped	Properly paired		
At	Oases	18.96	53.15	65.72	94.27	5.57	23 457	97.18	90.33	2.39	
	DRAP Oases	9.90	89.86	72.38	39.38	59.22	20 895	96.53	90.21	33.16	
	Trinity	38.97	45.40	40.32	81.09	17.69	20 290	93.81	85.78	10.04	
	DRAP Trinity	13.89	85.91	55.51	69.85	29.64	17 916	92.99	85.44	24.77	
Bt	Oases	36.07	56.48	28.29	93.33	6.61	10 560	90.53	87.20	2.71	
	DRAP Oases	32.59	67.13	25.70	67.63	32.05	10 456	91.03	88.59	23.30	
	Trinity	64.13	32.18	15.33	89.48	10.35	10 313	92.18	86.66	4.99	
	DRAP Trinity	38.55	61.23	24.86	79.95	19.91	10 144	91.03	85.97	13.51	
Dm	Oases	46.19	48.67	20.27	96.43	3.39	6 873	92.86	83.24	2.21	
	DRAP Oases	48.80	51.11	31.45	70.30	29.25	6 731	92.02	82.21	41.17	
	Trinity	67.53	27.89	18.49	89.63	9.42	6 494	93.24	85.07	17.56	
	DRAP Trinity	45.94	53.99	32.23	77.76	21.93	6 358	85.77	78.09	34.23	
Dr	Oases	56.81	40.39	23.37	97.98	2.01	15 186	85.73	75.16	0.67	
	DRAP Oases	40.20	59.65	33.43	70.89	28.84	14 901	88.26	82.84	25.19	
	Trinity	66.76	32.43	9.79	92.34	7.58	10 734	84.11	75.70	5.81	
	DRAP Trinity	39.74	60.19	20.16	82.44	17.44	11 272	81.33	75.43	18.25	
Ds	Oases	24.52	72.43	41.60	89.47	10.47	14 929	83.62	74.34	8.56	
	DRAP Oases	12.80	87.11	53.73	35.56	64.19	14 913	90.32	88.22	59.08	
	Trinity	37.72	60.46	30.29	67.37	32.26	14 394	88.79	85.37	38.77	
	DRAP Trinity	22.85	77.11	37.65	57.53	42.33	14 364	88.28	85.59	50.51	
Hs	Oases	44.51	49.94	21.18	93.04	6.93	7 554	88.30	NA	NA	
	DRAP Oases	46.95	52.91	20.06	77.28	22.64	7 463	86.90	NA	NA	
	Trinity	69.02	28.13	11.70	88.53	11.35	7 199	86.76	NA	NA	
	DRAP Trinity	55.48	44.41	16.07	83.46	16.48	7 124	84.08	NA	NA	
Notes.

Bold values are “best in class” values between raw and DRAP assemblies.

Table 4 Pooled samples vs meta-assembly strategies on the Danio rerio multi samples dataset (Dd)).

Assembly strategy		Pooled Oases	Meta Oases	Pooled Trinity	Meta Trinity	
Compaction	
Nb seq		42,726	43,049	62,327	65 271	
N50 (nt)		3,565	3,379	2,027	2 237	
L50 (nt)		10,409	9,259	14,956	13,106	
Sum (nt)		114,371,598	99,928,206	94,993,910	98,421,439	
Median length (nt)		2,182	1,766	1,217	1,052	
Contigs with multi-ORF (%)		0.33	0.50	0.13	0.17	
Contigs with multi-prot (%)		1.39	1.73	0.64	0.95	
Correction	
Chimeric contigs (%)		0.11	0.21	0.00	0.00	
Contigs with bias* (%)		75.19	68.00	58.79	61.88	
Validation	
% contigs by ORF count	0	24.79	38.77	37.24	50.63	
1	74.88	60.72	62.63	49.20	
Contigs with complete ORF (%)	61.84	46.36	38.80	31.55	
% contigs by proteins count	0	58.52	57.15	75.23	72.02	
1	40.09	41.13	24.13	27.03	
Nb reference proteins aligned	32,367	35,432	26,041	33,385	
Reads mapping (%)	Mapped	87.38	87.57	77.82	85.19	
Properly paired	78.88	80.13	70.13	77.30	
TransRate score * 100		28.66	29.49	17.97	23.36	
Notes.

* contigs with consensus variations corrected by DRAP.

Bold values are “best in class” values between raw and DRAP assemblies.

Contig set compaction

The improvement in compactness is measured by three criteria. The first is the number of assembled contigs presented in Fig. 4. The differences between raw Oases and Trinity assemblies and DRAP assemblies are very significant ranging from 1.3 fold to 15 fold. The impact of DRAP on Oases assemblies (from 3.4 to 15 fold) is much more significant than on Trinity assemblies (from 1.3 to 2,2 fold). Oases multi-k assembly strategy generates a lot of redundant contigs which are not removed at the internal Oases merge step. The second criterion is the percentage of inclusions, i.e., contigs which are part of longer ones. Oases and Trinity inclusion rate range respectively from 55 to 75% and from 2.3 to 5.5% (Table 2). Because of its inclusion removal step this rate is null for DRAP assemblies. The last compaction criteria presented here is the total number of nucleotides in the contigs. The ratios between raw and DRAP assembly sizes for Oases and Trinity range respectively from 3.4 to 14.8 fold and from 1.1 and 2.6 fold (Table 2). All these metrics show that DRAP produces less contigs with less redundancy resulting in an assembly with a much smaller total size.

Figure 4 Number of contigs.

The figure shows for the different assemblers (Oases, DRAP Oases, Trinity, DRAP Trinity) the number of contigs produced for each dataset.

Another metric that can be negatively correlated to compactness, but has to be taken into account, is the number of multi-ORF contigs found in the assemblies. The ratios of multi-ORF contigs found between raw and DRAP assemblies range from 11 and 116 folds (Table 2). DRAP multi-transcript chimera splitting procedure improves significantly this criterion.

In order to check if the compaction step only selects one isoform per gene, we compared the number of genes with several transcripts aligning on different contigs before and after DRAP. A transcript is linked to a contig if its best blat hit has over 90% query identity and 90% query coverage. The test has been performed on the Dr and the Ds datasets assembled with Oases and Trinity. The number of alternative spliced isoforms decreases more, with or without DRAP, in the Oases than in the Trinity assemblies (Table 5). This reduction is of 69% and 23% in the real dataset (Dr) and 83% and 18% in the simulated dataset for Oases and Trinity respectively. However, the spliced forms reduction does not impact the gene representation in the compacted sets (Table 5). Remarkably, the gene representation is increased for the real dataset when processed with DRAP Oases. This results from the different merging strategies used by Oases and DRAP Oases. Using TGICL, DRAP is able, in some cases, to correctly merge gene parts which have been generated by the Oases multi-k assemblies and this more efficiently than the build-in Oases merge procedure.

Table 5 Compaction vs gene representation on Danio rerio simulated dataset (Ds) and Danio rerio dataset (Dr).

Dataset	Assembly	Nb seq	All genes	Multi-isoform genes	Raw/DRAP assemblies	
					All genes (%)	Multi-isoform genes (%)	
Ds	Raw Oases	131,982	14,396	3,593	−1.74	−82.99	
	DRAP Oases	21,191	14,145	611	
	Raw Trinity	40,335	12,457	1,792	−2.04	−17.97	
	DRAP Trinity	31,113	12,203	1,470	
Dr	Raw Oases	702,640	11,613	2,177	+10.40	−69.09	
	DRAP Oases	46,831	12,821	673	
	Raw Trinity	126,210	8,310	801	−2.33	−22.60	
	DRAP Trinity	58,114	8,116	620	
Notes.

Bold values are “best in class” values between raw and DRAP assemblies.

Contig set corrections

DRAP corrects contigs in two ways: removing self-chimera and rectifying consensus substitutions, insertions and deletions when the consensus does not represent the major allele at the position in the read re-alignment file. Self-chimeras appear in Oases and Trinity contig sets at rate ranging respectively from 0.11 to 1.39 and from 0.09 to 0.56%. In DRAP, the corresponding figures drop to 0.01 to 0.16 and 0.00 to 0.01%. Concerning consensus correction only five datasets can be taken into account i.e., At, Bt, Dm, Ds and Hs. Dr Oases assembly generates such a large number of contigs and total length that it decreases significantly the average coverage and therefore limits the number of positions for which the correction can be made. As shown in Fig. 5 and Table S2, the Dr dataset is an outlier concerning this criteria. Regarding the five other datasets raw versus DRAP correction rates range from 1.7 to 18.6 for insertions, 3.1 to 27.1 for deletions and 2.7 to 14.1 for substitutions. DRAP correction steps lowers significantly the number of positions for which the consensus does not correspond to the major allele found in the alignment. In order to check the positive impact of the correction step, the Danio rerio reference proteome has been aligned to the simulated dataset (Ds) contigs before and after correction. 94.5% of DRAP Oases contigs and 86.2% of DRAP Trinity contigs which have been corrected, have improved alignment scores (Data S1 section “Contig set correction step assessment”).

Figure 5 Consensus error rates.

(A) presents the ratio of the global error rates between raw and DRAP assemblies for each dataset (data from Table 2 colum 12). (B), (C) and (D) present the ratio of the error rates respectively for substitution, insertions and deletions between raw and DRAP assemblies for each dataset (data from Table S2).

Figure 6 Reads re-alignment rates.

(A) and (B) show respectively the alignment rates for reads and read pairs for the four assemblies of each dataset.

Assembly quality assessment

The two previous parts have shown the beneficial impacts of DRAP on the assembly compactness and error rates but this should not impair quality metrics such as read and read pairs alignment rates, number of ORFs, complete ORFs found in the contigs, number of proteins of the known proteome mapped on the contigs or TransRate marks.

Read and read pair alignment rates differences between raw and DRAP assemblies are usually very low, between 1 and 2% and can sometimes be in favor of DRAP (Fig. 6) . In our test sets, the difference is significant (7.5%) for Dm when comparing Trinity to DRAP Trinity. This comes from the removal by DRAP of a highly expressed transcript (Ensembl: FBtr0100888 mitochondrial large ribosomal RNA) because that does not fulfill the criteria of having at least one 200 base pairs long ORF despite having over 11M reads aligned on the corresponding contig in the Trinity assembly. DRAP Oases assembly was not impacted because it builds a longer contig for this transcript with a long enough ORF to be selected in the additional part.

The reference proteome has been aligned on the contigs and matches with over 80% identity and 80% protein coverage have been counted (Fig. 7). These figures give a good overview of the amount of well-reconstructed proteins in the contig sets. For all datasets except one (At) the number of proteins are very close between raw and DRAP results. For this At dataset the difference is of 12.2% for Oases and 13.2% for Trinity. This is due to the FPKM filtering step performed by DRAP and the expression profile of this dataset that mixes different tissues (root, shoot and flower) and conditions (full nutrition and starvation). Contigs corresponding to low expression in one condition do not have sufficient overall expression to pass DRAP expression filter threshold and are therefore eliminated from the final set. Mixed libraries can benefit from the meta-assembly approach presented in the next section.

TransRate global scores (Fig. 8) are much higher for DRAP assemblies compared to raw ones. This comes from the compaction performed by DRAP and the limited impact it has on the read alignment rate.

DRAP has limited negative effect on the assembly quality metrics, and sometimes even improves some of them. Some cases in which multiple libraries are mixed with very distinct conditions can affect the results and it is good practice to systematically compare raw and DRAP assemblies. It is also to be noticed that Oases multi-k strategy outperforms Trinity for all datasets regarding the number of well-reconstructed proteins.

Pooled versus meta-assembly strategies

In the previous sections we compared results from raw and DRAP assemblies. This section compares results from pooled versus meta-assembly strategies both using the DRAP assembly pipeline (Table 4). Because of the read re-alignment filtering thresholds used in DRAP, we expect different metrics between a pooled assembly and merged per sample assembly (meta-assembly). DRAP includes the runMeta workflow, which performs this task.

Figure 7 Proteins realignment rates.

The figure shows the number of proteins which have been aligned on the contig sets with more than 80% identity and 80% coverage for each assembler and dataset.

Figure 8 TransRate scores.

The figure presents the TransRate scores of the four assemblers for each dataset.

Differences in compaction and correction are more important between Trinity and Oases than between pooled versus meta-assembly. Pooled assemblies collect significantly worse results for the number of reference proteins and number of read pairs aligned on the contigs. This comes from the filtering strategy which eliminates low-expressed contigs of a given condition when merging all the samples but will keep these contigs in a per sample assembly and meta-assembly strategy. Therefore, we recommend using runMeta when the assembly input samples mix distinct conditions with specific and variable expression patterns.

Assemblies fidelity check using simulated reads

The simulation process links each read with its transcript of origin. With this information it is possible to link contigs and transcripts. Here, the transcript-contig link was calculated using exon content and order in both sets (method explained in Data S1). The results presented in Table 6 first shows that the assembly process loses between 15.76 and 19.97% of the exons compared to the initial transcript set. This loss is close to 22% for all assemblies when the exon order is taken into account. As shown in Fig. 9, this is mainly the case for transcripts with low read coverage. The figures show once more that DRAP has a very limited negative impact on number of retrieved exons in correct order.

Table 6 Structure validation on Danio rerio simulated dataset (Ds).

Assembly	Retrieved exons	Exons in Right contig	Exons in Right order	Contigs with More than 1 gene	Max number Of genes by contig	
Real assembly	99.81%	99.81%	99.50%	0.16% (46)	5	
Raw Oases	80.03%	77.83%	77.61%	2.77% (537)	221	
DRAP Oases	80.21%	77.54%	77.29%	4.13% (671)	203	
Raw Trinity	84.24%	77.30%	77.10%	3.65% (717)	339	
DRAP Trinity	83.30%	76.65%	76.47%	3.17% (602)	327	
Notes.

Bold values are “best in class” values between raw and DRAP assemblies.

Figure 9 Gene reconstruction versus expression depth using simulated reads.

The figure presents the proportion of correctly build transcripts (method presented in Data S1 section “Contig validation using exon re-alignment and order checking”) versus the read count per transcript.

Table 6 shows the number of contigs linked to more than one gene. DRAP compaction and ORF splitting feature could have an antagonist impact for this criteria. But depending on the assembler, the figures are in favor or not of DRAP.

Table 6 also presents the maximum number of genes linked to a single contig. These clusters correspond to zink finger gene family members which have been assembled as single contig. Between 92.3 and 93.7% of the clustered transcripts belong to this family. De novo assembly tools are not able to distinguish transcript originating from different gene when the nucleotide content is highly similar.

Conclusion

Different software packages are available to assemble de novo transcriptomes from short reads. Trinity and Oases are commonly used packages which produce good quality references. DRAP assembly pipeline is able to compact and correct contig sets with usually very low quality loss. As no package out performs the others in all cases, producing different assemblies and comparing their metrics is a good general practice.

Supplemental Information

Data S1 Additional descriptions and validations

Click here for additional data file.

Table S1 Default command lines of DRAP third party tools

Click here for additional data file.

Table S2 Details about consensus error rates

Click here for additional data file.

Abbreviations

RMP Reads per million

ORF Open reading frame

TSA Transcript sequences archive

NCBI National Center for Bio-Informatics

DRAP De novo Rna-seq Assembly Pipeline

PNG, GIF, PDF or SVG Image file formats.

Additional Information and Declarations

Competing Interests

Author Contributions

Data Availability

The authors declare there are no competing interests.

Cédric Cabau conceived and designed the experiments, performed the experiments, wrote the paper, prepared figures and/or tables, reviewed drafts of the paper.

Frédéric Escudié performed the experiments, wrote the paper, prepared figures and/or tables, reviewed drafts of the paper.

Anis Djari performed the experiments.

Yann Guiguen and Julien Bobe analyzed the data.

Christophe Klopp conceived and designed the experiments, wrote the paper, reviewed drafts of the paper.

The following information was supplied regarding data availability:

De novo RNA-seq Assembly Pipeline: http://www.sigenae.org/drap.

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
