# Peer review of "Compacting and correcting Trinity and Oases RNA-Seq de novo assemblies"

_PeerJ, doi:10.7717/peerj.2988_

## Round 0.1 · original submission · Major Revisions

Dear authors,

Thank you for submitting your interesting article to PeerJ. Please excuse the extended turnaround time - I had some difficulties in securing reviewers. I hope their comments will be useful for your revision. Can you please pay in particular also attention to the practical comments on software.

Best wishes, Alice

Reviewer 1 ·

Basic reporting

1. There is for no reason the number 2 at the end of page 42 and 44.
2. It think labels in the table headers should not wrap (e.g.: "Dataset" in table 3).
3. What does "Contigs with bias" in table 2,3 and 4 mean? I could not find an explanation for this column.
4. In table 3 and 4 one column is labeled with "score * 100". I think score means the Transrate score. If this is the case then I think it should be labeled as Transrate score for clarity.

Experimental design

Although the “runCheck” functionality for testing the availability of all dependencies is quite useful, it is not enough to allow an easy installation.
I tried to follow the installation guide (http://www.sigenae.org/drap/install.html), but I could not
install DRAP on an Ubuntu 14.04 operating system because of the following reasons:

1. The tools “dc” and “cutadapt” are used in DRAP but are not listed as dependencies.

2. The python script Oasesv2.0.4BestTransChooser.py tries to call the interpreter with
“#!/bin/env python”. This is not directly supported by all unix systems. I fixed this by creating a symlink (ln -s /usr/bin/env /bin/env) on my system.

3. Even after installing and fixing the issues mentioned in point 1 and 2, I was not able to run the tutorial (http://www.sigenae.org/drap/quick_start.html). Error log:

Traceback (most recent call last):
File "/vol/scratch/drap-v1.8/bin/Oasesv2.0.4BestTransChooser.py", line 251, in <module>
ChosenNucSeqs.write(OutString+'\n')
NameError: name 'OutString' is not defined

Since DRAP has 22 dependencies (with dc and cutadapt) and uses shell, perl and python for scripting, I think it is quite difficult to install.

In order to simplify the installation process I suggest to package DRAP and all it's dependencies in a Docker (https://www.docker.com/) container. This would reduce the installation process to just two or three commands. Bioinformatic community projects like BioBoxes (http://bioboxes.org/) are already using Docker for tools like short read assemblers: e.g: https://github.com/bioboxes/velvet .

Validity of the findings

The authors are using the N50 metric to show how compact the assemblies are.
But the N50 metric does not state whether all the contigs used, represent the transcriptome. There could be contigs included that do not align to the reference.
Therefore I suggest to use NA50 metric which uses the alignment of the contigs against the reference.
NA50 is one of the outputs of the rnaQuast assembly evaluation tool (Bushmanova, E et al. ‎2016).

·

Basic reporting

DRAP is a software package for cleaning de novo RNAseq assemblies that
post-processes and combines Trinity and Oases assemblies. The primary
goal of DRAP seems to be to address the challenges of spurious creation
of transcripts due to a variety of (well known) RNAseq problems. A
secondary goal is to combine multiple RNAseq data assemblies.

This is a pretty decent paper in the often challenging area of RNAseq
assembly and analysis. The authors have done a nice set of benchmarks
and while I remain skeptical of some of their heuristics they are well
presented and argued.

My main comment is that the paper layout does not make it clear to me
which of the short-read RNAseq problems presented (starting line 45,
and continuing through line 96 or later) can and are addressed by
which of the steps in DRAP.

I also find the bar chart presentation (figures 1-5) of the data to
be completely impenetrable. I struggle with presenting lots of
numbers in some of my papers too, and don't have a clear suggestion
for the authors, but a table with numbers would be an improvement
already!

The paper writing needs to be proofread for English. There are many minor
grammatical issues that get in the way of understanding.

Experimental design

The authors used simulated data to test for preservation of
alternative splicing (line 255) and exon content and order (line
325). I expected to see similar assessments for matrices of table 2
and 3 and more importantly for assessment of contig set corrections.

Isn't #6 (line 54) the same as #2?

Line 72 - is there a citation for this (cite @6)? (Not necessary if
none exists, but I would like to know of one :)

Line 88 - is there a citation for this statement? The closest one I know
of is in the SOAPdenovotrans paper.

Line 118 - Trinity and other transcriptome assembler pipelines use
abundance normalization (in silico normalization or digital
normalization) to deal with combining reads from many large data sets.
Do the authors have a sense for whether runMeta is going to perform
better or worse on combining multiple assemblies than normalizing
and doing a global assembly that way?

Line 291, why would reference proteome with 80% identity be used? I would
expect whatever matches there were to be 100% identity, while allowing
for mismatches due to exons and UTRs. But there shouldn't be any difference
apart from that.

The most serious and intuitive drawback of 'Contig set compaction' in
RNAseq assembly is the elimination of the true alternative splicing
events; however the authors did not investigate this enough. They did
one experiment (line 255), which showed that 82% of genes with
multiple isoforms still have more than one isoform. They did not
report the reduction of splicing rate per gene. Also they did not
show how many alternatively spliced isoforms were recovered by the raw
assembly. Also, why did they do this with Trinity only and not Oases
where one would expect extreme loss of alternative spliced isoforms
because of the added compaction step?

Are consensus error rates corrected for the no of transcripts per
assembly? If not, the raw assemblies which have much more transcripts
must incorporate more errors. Also, you are calculating these error
rates with the same approach you use to correct them in the DRAP
pipeline. So there is no way to figure out if this approach itself is
introducing errors into the assembly. I think the simulated data is
a much more fair way to judge the correction algorithm. Another way
to assess the effect of this algorithm is to assess the conservation
scores of the changed bases or to show the effect on ORFs.

It seems that the pipeline performs filtration based on ORF length
(line 288) and transcript length (Supplementary figure 1) but the
methods section does not count these in the approaches used in Contig
set compaction!

One of the matrices that you used in the tables was
'Contigs with Bias' but you did not elaborate on
this in the text; please fix.

How did the authors define the cutoffs for self-chimera definition?

Is the error correction algorithm a home made script? If yes, which script
is doing that?

Validity of the findings

Addressed elsewhere.

Additional comments

Minor fixes:

Line 30, 1,3 => 1.3

Line 38, ease => easy

Line 59, proven => been proven

Line 62: 'but also chimeras'. 'But' does not fit.

Line 74: 'contigs contain local biological variations'. What do you mean by lo\
cal? Do you mean sample-specific?

Line 83: 'with a de Bruijn graph one'. Do you mean single Bruijn graph?

Line 117-121: You mention runAssessment first then runMeta but the sequence of \
the figures (S2 and S3) is in the opposite order.

Line 226: 'The last dataset'. This refers to the Ds dataset but actually you ar\
e talking about the Dd dataset

Line 241, 3,4 => 3.4

Line 242, 1,3 => 1.3

Line 284: You should refer to figure 3 not figure 4

Line 291: You should refer to figure 4

Installation and software:

The software is available from an in-house SVN server. Please post it
to a public site (preferably with a DOI) so that the version discussed
in this paper is available independently of the authors' institution;
most Web sites and servers disappear within a year or two. I suggest
using Zenodo or Figshare, and perhaps mirroring it on github (which
will work with Zenodo). (Please address this.)

DRAP is licensed under GPL v3, so a full OSS license - excellent! I'd
suggest mentioning this in the abstract.

The list of dependencies was too long and specialized for me to
consider installing it. While I don't think improving this should be
required for publication, this is going to be a significant barrier to
use. I would suggest providing instructions using brew (for OS X) and
for apt-get (Ubuntu/Debian) or rpm (RedHat). I would be happy to try
out the software if I could install the dependencies with a few
commands.

The documentation seems decent and provides example command lines and
test data sets - great!

I note that DRAP suggests installing an old version of Trinity from
2014, trinityrnaseq-r2014-07-17; why? There's a release out Mar 17,
2016 (v2.2.0) and the paper should use a more recent version of
Trinity for its benchmarks. (Please address this.)

What is the execution requirement for DRAP in terms of memory and disk
space? I understand that this will necessarily be on top of whatever
Trinity and Oases require; is it significantly greater? RAM is a
big problem for large data sets, typically. It would be good to see
some basic runtime figures just to get a sense of what is required to
run DRAP. (Please address this.)
* * *
Note, this review was co-authored by Dr. Tamer Mansour.

·

Basic reporting

In the supplement the authors show two figures (S1 and S2) that detail the different steps as part of their pipeline. First of all the figures are well made and nicely show the workflow. Second of all, I find it hard to follow all the different steps of their pipeline right now as these are scattered across many pages in the Methods part. Therefore I would like the authors to move them to the main paper.


Currently all figure and table legends are not comprehensive and should be expanded to explain the metrics used (for example the term scoring should be referred to as Transrate score). For example in table 4 it is not clear on which dataset(s) the strategies were compared.

Experimental design

No comments

Validity of the findings

No comments

Additional comments

The authors present a post-processing pipeline for the Trinity and Oases assemblers.
The authors have investigated several datasets from different species. I find that the authors have some interesting ideas for improvement of the downstream results of assemblers. The authors show that their DRAP pipeline generally leads to few losses in terms of transcripts with ORFs and shows good read realignment rates.

Additional comments:
The authors use the prediction of ORFs in the assembled transcript as one quality filter. While this is a sensible idea, it may also be disadvantageous if there is interest in noncoding RNA transcripts. This should be clearly discussed, and is also one of the reasons that I favor to have the pipeline workflow in the main paper.

The authors use realignment rates before using DRAP and after, as one quality metric. I think they should discuss, that this is a little biased, as they correct contig sequences through the investigation of read alignments. Therefore, this contig correction alone, should normally result in an improvement of the realignment statistic. However, because they also remove genuine sequences they loose reads that cannot align anymore.
In this light, I would find it interesting to investigate in more detail the Ds dataset with Oases, where the realignment rates increase after DRAP by ~7%. Probably its due to the fact that they loose almost no genuine sequences that produce ORFs.

In the introduction the authors already cite a few works that address the problem of compacting assemblies. I would like the authors to add two additional works:
1: Corset: enabling differential gene expression analysis for de novoassembled transcriptomes. Genome Biology 2014. They have a nice method for clustering assemblies using replicate information.
2: Informed kmer selection for de novo transcriptome assembly, Bioinformatics 2016. In our paper about multi-k assembly analysis (not only) with Oases, we show that what we term single-k clustered transcripts, can be easily removed, as these often represent miss-assemblies, without significant impact on the ability to find true transcripts.

I would like that the authors present runtime results of their pipeline on the diverse datasets.

Kind regards,
Marcel Schulz

---

## Round 0.2 · accepted · Accept

Thank you for providing the revised version and happy new year.